# Electrospun Mats Based on PVA/NaDDBS/CNx Nanocomposite for Electrochemical Sensing

**DOI:** 10.3390/ma14216664

**Published:** 2021-11-05

**Authors:** Paloma Vilchis-León, Josué Hérnandez-Varela, José Jorge Chanona-Pérez, Raul Borja Urby, Rodolfo Estrada Guerrero

**Affiliations:** 1Laboratorio de Nanociencia y Nanotecnología, Departamento de Física y Matemáticas, Universidad Iberoamericana, Ciudad de México 01219, Mexico; roldolfo.estrada@ibero.mx; 2Laboratorio de Micro y Nano-Biotecnología, Departamento de Ingeniería Bioquímica, Escuela Nacional de Ciencias Biológicas, Instituto Politécnico Nacional, Ciudad de México 07738, Mexico; jhernandezv1717@alumno.ipn.mx (J.H.-V.); jorge_chanona@hotmail.com (J.J.C.-P.); 3Center of Micro and Nanotechnology of IPN, Ciudad de México 07738, Mexico; rborjau@ipn.mx

**Keywords:** PVA, nanofibers, CNx, electrospinning, electrochemical

## Abstract

This study presents a nanocomposite developed with PVA, multiwall carbon nanotubes (CNTs) doped with nitrogen, and NaDDBS, which change the electrical properties of the polymer and its viscosity to be used in electrospinning process for obtaining mats of nano/macro fibers. The proposed nanocomposite was characterized using Fourier transform-infrared and Raman spectroscopy techniques, confirming the presence of the CNxs immersed in the polymer. High-resolution transmission electron microscopy was used to obtain the micrographs that showed the characteristic interplanar distances of the multiwall CNT in the polymeric matrix, with values of 3.63 Å. Finally, the CNx mats were exposed to various aqueous solutions in a potentiostat to demonstrate the effectiveness of the nanofibers for electrochemical analysis. The CNx-induced changes in the electrical properties of the polymer were identified using cyclic voltammograms, while the electrochemical analysis revealed supercapacitor behavior.

## 1. Introduction

Nanocomposites based on biopolymers have made an impact on the scientific community due to environmental concerns. Poly (vinyl alcohol) also known as PVA is an ecofriendly, biocompatible and hydrophilic polymer; it has many hydroxyl groups, strong covalent bonds, and is a semicrystalline polymer. These characteristics make PVA ideal for use in colloidal solutions and novel nanocomposites [1].

As is well known, MCNT improves the mechanical and electrical properties when combined with other materials such as polymers. However, the presence of the Van der Wall’s interactions between MCNT and the polymer matrix could affect the performance of the novel nanocomposite [2]. This can be avoided through the use of a surfactant [3] to ensure the dispersion of the MCNT in combination with some doped options like multiwalled carbon nanotubes doped with N [4,5,6].

These nanocomposites are used in an electrospinning process to obtain fibers of different sizes. The electrospinning process is an inexpensive method to obtain micro- and nanofibers [7]. A sufficiently high voltage is applied to a liquid to initiate electrospinning, which begins to accumulate charge, forming droplets until it reaches a conical form known as a Taylor cone. The size of the fiber depends on the distance between the collector plate and the Taylor cone, the voltage, and the viscosity of the liquid [8].

The electrospinning process provides a low-cost alternative for fabricating biosensors, especially those focused on electrochemical sensing. The electrochemical biosensors have been demonstrated to have application in food technology, the biomedical field, water filtration, and more recently, in the recognition of biomolecules.

The combination of the nanocomposite and the easy electrospinning process to fabricate mats of fibers present an alternative to commercial electrodes for electrochemical sensing. The modified PVA with different types of nanoparticles such as TiO_2_ [9], MnO_2_ [10], thermally reduced graphene [11] has demonstrated that the PVA-based nanocomposite is suitable for this type of recognition. The recognition is possible thanks to the modification of screen-printed electrodes with mats of nanofibers. This work presents an alternative to using a commercial electrode cover with nanofibers made of a nanocomposite of polyvinyl alcohol, surfactant, and multiwall carbon nanotubes doped with N, for further use in biological sensing.

## 2. Materials and Methods

### 2.1. Materials

The main components of the proposed nanocomposite, namely PVA (98% hydrolyzed, an average molecular weight of 72,000 g mol^−1^) and functionalized nitrogenate carbon nanotubes (CNx), were synthesized by a CVD process. Raw nitrogenate carbon nanotubes were used as reference material. Sodium dodecylbenzenesulfonate (NaDDBS), used as the surfactant, was purchased from Sigma Aldrich, CDMX, México. Distilled water was used for all processes.

### 2.2. Synthesis of CNx

CNx was produced using a CVD process with 200 mL of the solution containing 5 wt% of FeCp_2_ (ferrocene) in C_7_H_9_N (benzylamine) [12,13]. When the furnace reached 850 °C, argon (Ar) gas was at 0.1 SLPM (standard liter per minute) and 14.1 psi, at the desired temperature; the gas was changed to 2.1 SLPM and the sprayer containing the previous ferrocene and benzylamine solution was turned on. This setup was left for 30 min.

After 30 min, the sprayer was turned off, as also the furnace; during the cooling process, the gas flow was reduced to 0.1 SLPM. To avoid benzylamine contamination of the CNx, the extreme of the tube was cleaned with acetone. Finally, 0.5 g of CNx was obtained, and this synthesis was repeated twice to gather 1 g of the carbon nanotubes doped with nitrogen.

### 2.3. Purification of CNx

Purification is the process of eliminating the amorphous carbon from CNx. For this, a mixture of HNO_3_ and H_2_SO_4_ at 1:3 was stirred constantly at room temperature for five hours. To neutralize the acid a solution of NaOH, 1 mol was prepared with dH_2_O. Finally, the CNx was cleaned with ethanol, dH_2_O, and acetone for five cycles [14].

### 2.4. Electrospinning Procedures

Solutions of PVA and NaDDBS were prepared separately. The latter was prepared with 5 mL of dH_2_O, 50 mg of NaDDBS, and 1 g of CNx. The solution was sonicated and stirred for three cycles lasting 30 min each. For the PVA solution, a 10% (*v*/*v*) concentration was used. Both solutions were mixed on a hot plate while being stirred to avoid CNx aggregation and clusters of PVA, to obtain the required viscosity for electrospinning and were added to achieve an 8.695% *v*/*v* in the final solution.

Electrospinning was performed with a syringe pump (Kd Scientific, Holliston, MA, USA) in a 5 mL syringe, with a velocity of 0.5 µL/min. The distance between the collector plate and the syringe tip was 15 cm. The process was performed at 10 kV, using two power supplies connected in series. Nanofiber mats were obtained after 40 min of running the pump, after confirming that the viscosity remained constant and the tip residue-free. 

### 2.5. Fourier Transform-Infrared (FT-IR) Spectroscopy

To characterize the chemical groups in the polymer, surfactant, and CNx, FT-IR spectra were obtained using a Nicolet i10s FT-IR spectrometer (Thermo Scientific, Waltham, MA, USA). Spectra were obtained separately for CNx as a powder; one mat of nanofibers was made with the nanocomposite and another nanofiber mat with PVA alone. All spectra were recorded in the range of 4000–650 cm^−1^, and 20 scans were taken for four samples of each material. The spectra were recorded at a resolution of 4 cm^−1^ and normalized at 1030 cm^−1^.

### 2.6. Raman Spectroscopy

Raman spectroscopy was performed using a LabRam HR 800 Raman spectrometer (Horiba Jobin Yvon, Kyoto, Japan) coupled to an Olympus BX 41 microscope with a 100× objective. Raman spectra were recorded using a 600 lines/mm grating and a 653 nm emission laser. The spectral resolution was approximately 4 cm^−1^. The measurements were conducted spanning the 100–3200 cm^−1^ wave number and using exposure times of 8–10 s, in the range of 19 °C of temperature.

### 2.7. Electron Microscopy

For scanning electron microscopy (SEM), small samples of the mats were cut and examined using a Hitachi SU-3500 (Minato-Ku, Tokyo, Japan) instrument under high vacuum conditions. Images were acquired using the secondary electron detector. The samples were analyzed using slow frame rates of 3 and 15 Hz and a working distance of 6 mm. Further, SEM images were used to evaluate the average fiber diameter (AFD) by measuring each fiber size with the length tool of ImageJ software version 1.47 (http://imagej.nih.gov, accessed on 16 October 2021; National Institute of Health, Bethesda, MD, USA). The collected data were plotted as frequency histograms and adjusted to a Gaussian distribution function in SigmaPlot software v 12.0 (Systat Software Inc., San Jose, CA, USA). The goodness of fit of the models was evaluated by their coefficient of determination (R^2^). For high-resolution transmission electron microscopy (HR-TEM), the mats were scraped. Some of the nanofibers were wetted with a droplet of isopropyl alcohol, deposited onto a copper grid for transmission electron microscopy, and left to dry for 15 min. The TEM (JEOL, Peabody, MA, USA) was operated in bright-field mode at 80 kV to increase the contrast between CNx and the surrounding polymeric matrix [8,15]. Micrographs were analyzed in GMS 3 software package (Gatan Microscopy, Pleasanton, CA, USA) using the methodology proposed by Hernández-Varela [16]. Images were processed through fast Fourier transformation (FFT) for the crystalline regions (CR) of the sample. An inverse fast Fourier transformation (IFFT) was produced using the crystalline fringe of the reciprocal space from a particular mask, to produce a higher resolution representation of the interplanar distances. Finally, images were stored in TIFF format and used for discussion. 

### 2.8. Electrochemical Characterization

Electrochemical characterization of the fabricated PVA/CNT/NaDBBS nanofiber composite was performed using a potentiostat/galvanostat PGSTAT101 (Metrohm Autolab, Utrecht, Netherlands) connected to a PC with NOVA software used to control the potential, data acquisition, and treatment. The experiments were conducted using a conventional electrochemical cell with a three-electrode adaptation connector. For comparison, commercial electrodes (screen-printed electrodes) and a mat of nanofibers made only with PVA were used in the analysis. The electrodes were characterized using a solution composed of 0.1 mol L^−1^ potassium ferricyanide (K_3_Fe(CN)_6_) and potassium ferrocyanide (K_4_Fe(CN)_6_) in 1 mol L^−1^ KCl, as well as, 1 mol L^−1^ KCl as control electrolyte. The electrochemical window was established between −0.2 and 0.6 V and a 10-scan rate. All reactants were of analytical grade and had undergone no previous purification. The solutions were prepared in Milli-Q water (18.2 MΩ cm; Millipore Corporation-Merck, Burllington, MA, USA). 

## 3. Results

### 3.1. Effect of Purification in the CNx

Before using the CNx in the nanocomposite, a SEM micrograph was obtained, to compare the purification of the forest of CNx. Figure 1a shows raw CNx, while Figure 1b shows the CNx after the purification process. Although they appear to be not completely dispersed, they are not in groups.

### 3.2. Characterization of CNx

Figure 2 presents a Raman spectrum, acquired at 653 nm excitation, collected from raw nitrogenate carbon nanotubes (CNx) and functionalized nitrogenate carbon nanotubes (CNx-F) in powder form. The radial breathing modes (RBM) associated with large-diameter tubes are too weak to be observed in these spectra (100–200 cm^−1^) [17]. The D-band shows a high-intensity peak at 1340 cm^−1^, corresponding to the induced disorder in the carbon nanotubes, and the G-band shows a tangential peak at 1586 cm^−1^, which is related to the tangential E2g Raman active mode of graphite and caused by the tangential vibration of the two atoms in the graphene unit cell against each other [18]. In the second-order bands, a group of weak peaks can be observed between 2673 and 2955 cm^−1^, which correspond to the G′ and D + G modes. For the D- and G-bands, functionalization caused a relative increase in the height of the bands. This behavior was expected, as chemical functionalization-induced increases in the D-band intensity have been reported in previous studies [19].

### 3.3. Effect of Surfactant on the Dispersion of PVA/CNx/NaDBBS Composite

CNTs have a high surface tension due to Van der Waal’s interactions between them [15], provoking some aggregation problems due to the strength of these forces. However, some studies have shown that good CNx dispersion can be achieved by using a surfactant [3]. The surfactant reduces the surface tension of the CNx, thus preventing aggregation. In addition, sonication can be used to achieve good dispersion in the solution, while several studies have reported that using a group of surfactants produces good results [3,19]. In all cases, NaDDBS was selected as an effective dispersant because it contains hydrophilic and hydrophobic parts that cause a double reaction when they are in solution [20]. The hydrophobic part adsorbs on the CNx surface, while the hydrophilic part dissolves in the aqueous solution. Figure 3 shows a schematic representation of the interactions between CNxs, NaDDBS, and PVA in the final composite. The molecular chains of NaDDBS can be inserted between the disorganized CNx composite, creating a well-defined tridimensional grid when the CNxs are aligned and dispersed by the surfactant. Thus, by introducing a low-molecular-weight polymer such as PVA into the solution system, the PVA molecules can intercalate their structure in the CNx grid. Consequently, a final composite comprising a PVA/CNx/NaDDBS solution was obtained, suitable for producing electrospun nanofibers.

### 3.4. Chemical and Physical Characterization of PVA/CNT/NADBBS Composite

Electrospun PVA and PVA/CNx/NaDBBS were observed using SEM. Figure 4 shows the SEM images of PVA and PVA/CNx/NaDBBS after 40 min of electrospinning, revealing their respective fiber diameter distributions. As shown in Figure 4a, in the pure PVA sample, the average fiber diameter was found to be in the range of 640 nm, while in Figure 4b with 40 min of electrospinning, the average fiber diameter was 470 nm. From these micrographs, the role of the electrospinning time as a variable for achieving small nanofiber-based structures can be inferred. The reduced diameter of the nanofibers in the composite is attributed to the increased stretching of the fibers during electrospinning, caused by the increased charge due to the presence of conductive CNxs in the polymer solution. The diameters of the electrospun fibers (Figure 4c,d) can range from several microns to tens of nanometers. Together, small fiber diameters and a large aspect ratio lead to an extremely high surface-to-volume (weight) ratio, rendering the electrospun nanofibers desirable for many applications, such as sensing devices [21].

Next, FT-IR spectroscopy was used to assess the chemical groups in the polymers after electrospinning. Figure 5 shows the FT-IR spectra of PVA and PVA/CNT/NaDBBS after 40 min of electrospinning. In Figure 5, the major peaks in the FT-IR spectrum of PVA all relate to hydroxyl and acetate groups. More specifically, the broadband observed between 3400 and 3100 cm^−1^ is associated with the O–H stretching from inter- and intramolecular hydrogen bonds. The vibrational band observed between 2800 and 2980 cm^−1^ is the result of C–H stretching in alkyl groups, while the peaks between 1760 and 1510 cm^−1^ are due to the C=O and C–O stretching in the remaining acetate groups in PVA (owing to the saponification of PVA). Another characteristic peak of PVA below 1500 cm^−1^ corresponds to the C–C and C–O–C interactions in the polymeric matrix [21]. Other peaks at 823 cm^−1^ and 642 cm^−1^ as a result of the benzene substitutes and aromatic elements [22,23,24]. Since the normalization of the spectra at 1030 cm^−1^ is used to avoid false interpretation of the data, it seems that the dotted curve is a magnification of the solid curve. These results suggest an interaction between CNx and the polymer matrix with the O–H group and –NH_2_ groups involved in the composite [17,25].

Additionally, Raman spectroscopy was used to evaluate the presence of CNTs and track the interactions of the composite after the time taken to obtain the mats. Figure 6 presents the Raman spectra of PVA and PVA/CNx/NaDBBS after 40 min of electrospinning. To understand the effect of adding the synthesized nitrogenate CNTs on the internal structure of the composite, the Raman spectra of PVA with and without CNx are also shown in Figure 6. It can be seen that except for the intrinsic ns (CH_2_) stretch band at 2910 cm^−1^, other characteristic bands for pure PVA are observed at 1440 cm^−1^, 857 cm^−1^, 912 cm^−1^, and 480 cm^−1^ [17,22]. However, when CNT loading levels were detected according to the characteristic peaks of CNT tangential modes, the well-known D-band, G-band, and G′-band suffer some shifting to values of 1364 cm^−1^, 1602 cm^−1^, and 2720 cm^−1^, respectively [22,26]. These displacements occur due to the stronger attachments of the CNx onto the polymeric matrix of the PVA and the influence of the surfactant in the CNx dispersion.

On the other hand, some intrinsic factors, such as the polymer solution parameters (molecular weight, molecular weight distribution, electrical conductivity, surface tension, viscosity, and solvent type) and extrinsic factors, such as the operating parameters (electrical field, the distance from the nozzle tip and the collector, and the flow rate of the polymer) were evaluated to minimize the random errors in the production of the fibers; however, they were not included for discussion. Moreover, the ambient conditions were considered [4] to establish the best way to obtain the mats.

Finally, a high-resolution analysis using TEM images was made to evaluate the interaction between the polymer and CNx. Figure 7 shows an image of PVA/CNT/NaDBBS electrospun nanofibers and their high-resolution image analysis. Figure 7a presents the regular configuration of each nanofiber in polymer mats created with electrospinning. The nanofibers have sizes around 200 nm, as shown in SEM images; in contrast, Figure 7b shows the amplification of the inset selected in Figure 7a. As expected, some irregularities or amorphous regions (AR) are shown for the PVA/NaDBBS material. However, crystalline regions are observed (CR) in Figure 7b, attributed to bunches of nanotubes present in the intermolecular chain of PVA, provoking a regular surface morphology not observed in PVA. Using the selected region in Figure 7b (dashed square), an edge-on crystalline lamella was evaluated by applying an FFT. The crystalline region in the reciprocal space (inset, Figure 7b) was used to produce a mask and later on, produced an IFFT to measure the interplanar distance in the sample (Figure 7c). For this case study, the interplanar distances on CNx were measured with the software, and values of 3.63 ± 0.23 Å obtained. Similar results were found for interplanar distances of multiwalled carbon nanotubes using HR-TEM. Singh et al. [27] reported values between 3.8–3.2 Å when CNT diameter sizes change from 5 to 100 nm. 

### 3.5. Electrochemical Characterization

The current-potential characteristics experiments of three different samples were obtained using a commercial screen-printed electrode (solid line), with PVA (dotted line) and the nanocomposite (dashed line) shown in Figure 8. Cyclic voltammetry (CV) at 100 mV/s in two solutions used to characterize the electrodes was composed of 0.1 mol L^−1^ potassium ferricyanide (K_3_Fe (CN)_6_), 1 mol L^−1^ potassium ferrocyanide (K_4_Fe (CN)_6_), and 3 mol L^−1^ KCl. First, a commercial electrode (solid line) was used as a control response in the KCl solution at room temperature (Figure 8a); another electrode was used and put on one section of a small piece of PVA mat (dotted line). Finally, another electrode was covered with the nanocomposite mat (dashed line), in the same KCl solution. The same experiment was conducted for the electrolyte solution Fe (CN)_6_^−3^/Fe (CN)_6_^−4^ (Figure 8b).

Figure 7 shows the electrochemical responses for raw control electrodes, control electrodes with a mat of PVA and PVA with CNx, in different aqueous solutions. For a common electrolyte solution of KCl 1M (Figure 8a), the control electrode has almost zero response (solid line), and the electrode just with PVA (dotted line), increases the conductivity of the systems by introducing a small quantity of the surfactant that changes the electrical properties of the polymer. Nevertheless, it is important to control the addition of surfactant because high amounts thereof could reduce the viscosity of the solution almost to the level of water and could form micelles or aggregates. Finally, when CNx are added to the PVA mat, two characteristic peaks appear around +0.17 V and −0.02 V (dashed line) because of the redox process involved in the system. It is important to explain that both peaks are missed in the electrode with just PVA and raw electrode, but the intensity of the peaks is not significant to evaluate the redox process in the systems. 

Hence, a complex electrolyte solution based on potassium ferricyanide was used, as shown in Figure 8b, which presented high surface-sensitivity, with a noticeable response dependent upon the carbon-oxygen surface groups of the mats in the redox complex [7]. When the control electrode is evaluated (solid line), a classical response is presented in the electrochemical analysis, while the electrode with only PVA and the electrode with the PVA/CNx nanocomposite shows a reversible redox process. But the nanocomposite with PVA/CNx presents two peaks at +0.18 V and −0.05 V, which is a desirable value for electrode conduct for sensing biological analytes and it is a desired behavior for super-capacitors [24,25].

## 4. Conclusions

The applications of polymer nanocomposites have grown exponentially in medical and biological fields, tissue engineering, and rapid sensing. This work has presented a novel nanocomposite based on PVA/NaDDBS/CNx, which can be used to reinforce the commercial screen-printed electrode for electrochemical analysis. The Raman spectroscopy demonstrates the quality of the CNx and the presence in the polymer matrix. The FT-IR confirms the bonds between the hydroxyl and acetate groups characteristic of PVA, and peaks at 823 cm^−1^ and 642 cm^−1^ resulting from the benzene that was used in the CVD process synthesis of CNx. 

The TEM and SEM micrographs confirm the crystalline regions attributed to bunches of nanotubes present in the intermolecular chain of PVA. The measure of the interplanar distance was found to be 3.63 ± 0.23 Å. The stability of the developed nanocomposite in the electrochemical analysis was indicated by the probe peaks present at +0.18V and −0.05V, which is a desirable value for electrode conduct for sensing biological analytes.

These nanocomposite mats can be used as electrodes for electrochemical sensing. In addition, the nanofiber composite mats exhibit potential as supercapacitors and can be used as selective biosensors in Bio-MEMS for diagnostic purposes, and in a more present and urgent application as a filter or trap for biological pathogens.

## Figures and Tables

**Figure 1 materials-14-06664-f001:**
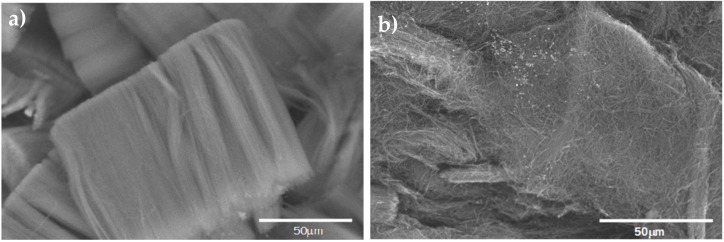
(**a**) CNx without purification, (**b**) CNx after treatment with acid for purification.

**Figure 2 materials-14-06664-f002:**
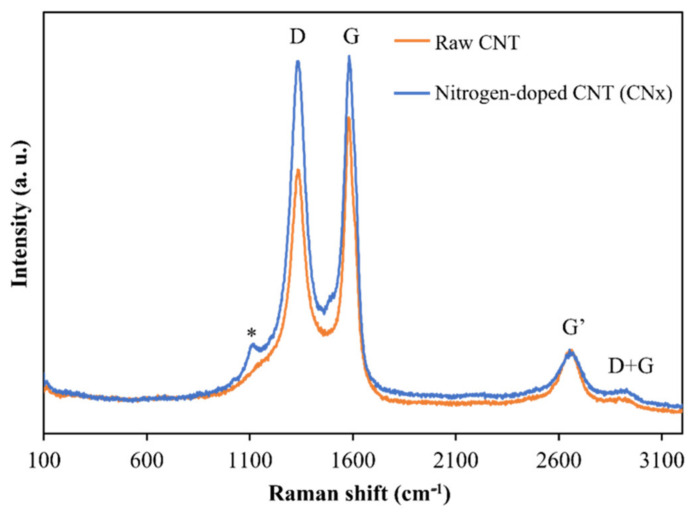
Raman spectra of raw and functionalized CNx.

**Figure 3 materials-14-06664-f003:**
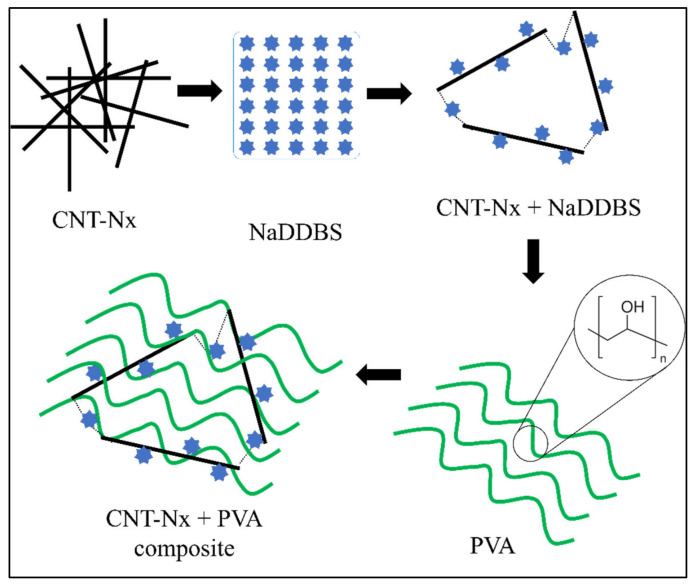
Schematic illustration of the interactions between CNT, NaDDBS, and PVA in the final composite.

**Figure 4 materials-14-06664-f004:**
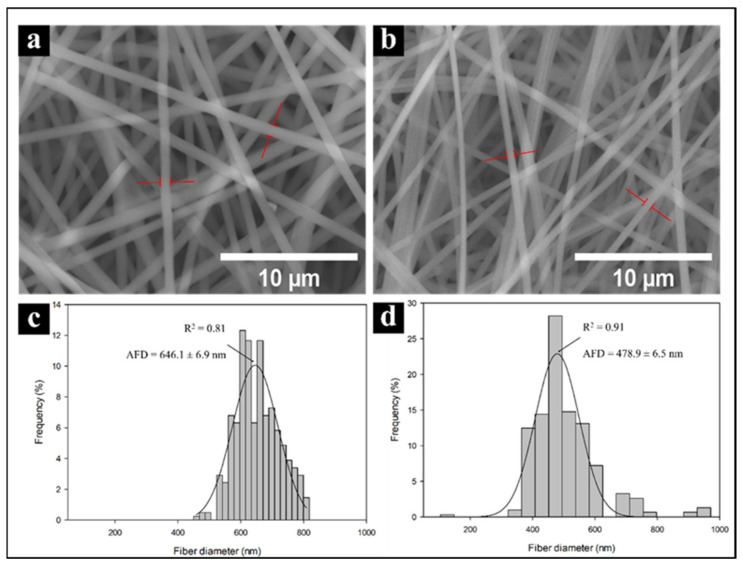
Scanning electron micrographs and histograms of fiber diameters for PVA nanofibers (**a**,**c**) and PVA + CNx nanofibers (**b**,**d**). AFD = average fiber diameters.

**Figure 5 materials-14-06664-f005:**
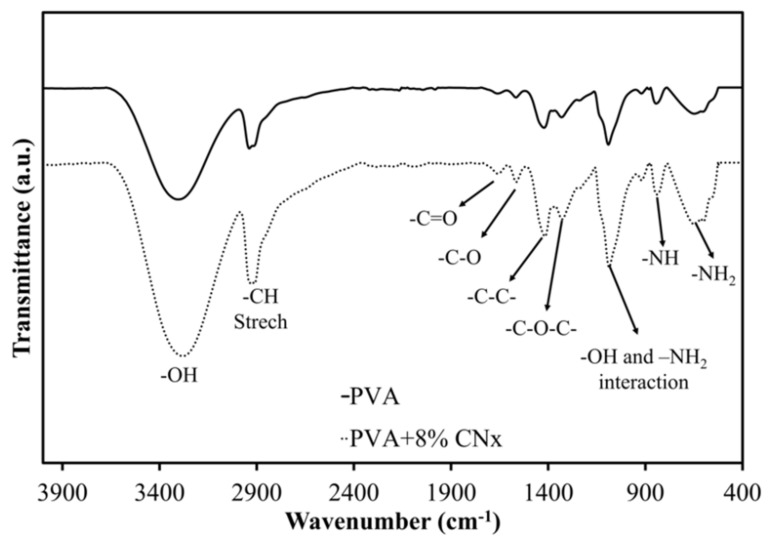
Fourier transform-infrared spectra highlighting the chemical groups in PVA (solid line) and PVA/CNT/NaDBBS (dotted line) after electrospinning.

**Figure 6 materials-14-06664-f006:**
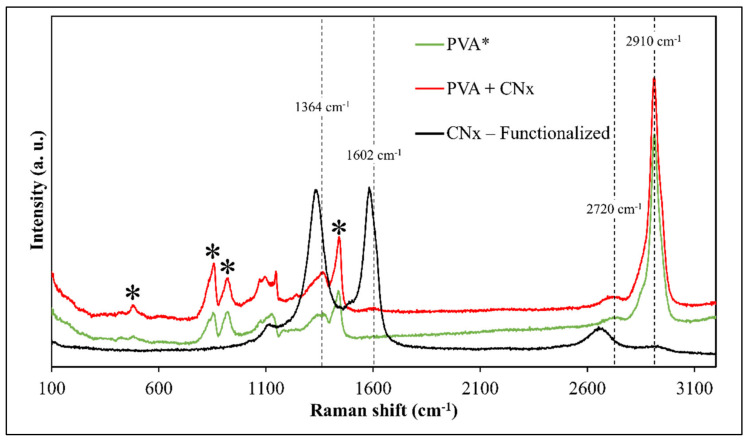
Raman spectra of PVA (green), PVA/CNx/NaDBBS (red), and CNx-functionalized (black). Black asterisks (*) represent the typical signal for PVA.

**Figure 7 materials-14-06664-f007:**
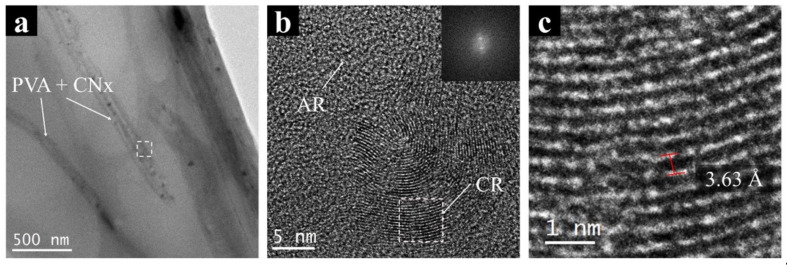
Transmission electron micrographs of (**a**) PVA + CNx nanofibers and (**b**) the magnification image of the nanocomposite showing their crystallographic structure. (**c**) Reconstruction of the FFT to IFFT showing the interplanar distances in CNx. AR: amorphous regions; CR: crystalline regions.

**Figure 8 materials-14-06664-f008:**
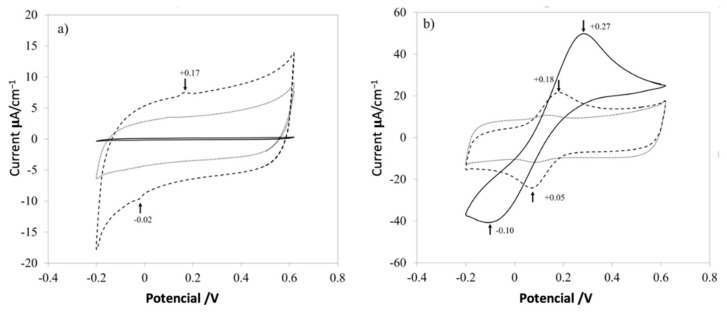
Electrochemical kinetics of PVA nanofibers on a screen-printed electrode. Cyclic voltammograms of the screen-printed electrode (solid line), pure PVA (dotted line), and PVA + CNx (dashed line), showing their electrochemical response to an aqueous solution of 1 M KCl (**a**) and Fe (CN)_6_^−3^ /Fe (CN)_6_^−4^ (**b**).

## Data Availability

The raw data supporting the conclusions of this article will be made available by the author without undue reservation.

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
