# Peer review of "Electrospun Mats Based on PVA/NaDDBS/CNx Nanocomposite for Electrochemical Sensing"

_materials, 2021, doi:10.3390/ma14216664_

Round 1
Reviewer 1 Report
This manuscript deals with preparation of electrospun PVA fibers with carbon nanotube and its application for electrochemical sensing. Generally, this is not a new approach, however, I consider the article is important from the practical point of view of the manufacturing of sensing materials, which is a current research issue.
- Please use a title that specifically describes your research. Its meaning is too comprehensive.
- The novelty of the work should be strengthened in introduction.
- The background and purpose in Introduction are not very clear. Please make those more clear.
- Please cite the latest bibliographies.
- The experiment is well-organized in detail. Readers will probably want to know about information of carbon nanotubes. Please provide the synthesis method for carbon nanotube, or refer to the paper providing the synthesis method of the carbon nanotube you used
- Can you prove images of the manufactured mat?
- In results, the effect of variation of CNx and NaDDBS loading on properties of the mats. Are you sure those manufacturing conditions are optimal?
- Please check line 292, XX mV/s
- Please provide a discussion of your research results, and conclusions from them.
Author Response
"Please see the attachment."

Reviewer 2 Report
In the manuscript entitled “Polymer Nanofiber Mats for Electrochemical Sensing” Paloma Vilchis-León et al. propose a nanocomposite system based on polyvinyl alcohol and nitrogen-doped MWCNTs as electrochemical sensor.
First of all, the manuscript title is too generic; it would be better to specify the nanocomposite composition.
In the abstract, the authors should eliminate the first part, and start with “This study presents…”; moreover, in the abstract the authors should briefly indicate the best obtained results, in this case the electrochemical properties of proposed nanocomposite system.
In the introduction, the authors should briefly describe PVA (polyvinyl alcohol) and its properties.
In the introduction, the authors should report the application, reported in literature, of PVA as sensing material, adding the best obtained results.
In the introduction, the authors should report the commonly used method, reported in literature, to prepare PVA nanofibers, and their application as sensing layer; moreover, the authors should report the hybrid polymer/MWCNTs systems reported in literature as electrochemical sensing layers. Therefore, the authors should better motivate the choice of use PVA in combination with MWCNTs, specifying the reason to use nitrogen-doped MWCNT (what does nitrogen-doped MWCNTs means? How is its chemical structure?!); which are the interactions between PVA and nitrogen-doped MWCNTs? Discuss.
In the experimental section, more details on nitrogen-doped MWCNTs should be added.
Considering the SEM reported images, it is evident that the dimension o PVA fibers (the length) is micrometer; therefore, it would be better refer to composite and not nanocomposite system.
For a correct comparison, it is necessary also report SEM and TEM images, FT-IR spectra and the electrochemical characterization of the reference nitrogen-doped MWCNTs.
For a better comprehension, the authors should also indicate which is the quantity of nitrogen-doped MWCNTs added to PVA, exploring also the effect of its various contents on the chemical, morphological and electrochemical properties of the composite.
The English style of the manuscript should be improved.
I can accept this manuscript with major revisions.
Author Response
"Please see the attachment."

Reviewer 3 Report
Dear authors I suggest you to revise your manuscript as concern:
Figura 4 - My opinion is that your interpretation of the signals "Other peaks at 823 cm-1(– 237 NH) and 642 cm-1 (–NH2) result from the interactions between CNx and PVA in the matrixat" does not convince me.
"3.4 Electrochemical Characterization"
You have to write the redox reactions you think are involved.
You have to make cyclic voltammetry at different rates.
If possible you have to calculate the electron charges that are involved in these redox process.
I give you these suggestions to allow you not only to show "curves" but to hypothize in your paper an interpretation of the redox processes you claim.
best regards
Author Response
Figura 4 - My opinion is that your interpretation of the signals "Other peaks at 823 cm-1(– 237 NH) and 642 cm-1 (–NH2) result from the interactions between CNx and PVA in the matrixat" does not convince me.
Yes, that peaks are the presence of the benzene for the synthesis of the CNx. In the new manuscript we correct the information.
Round 2
Reviewer 1 Report
Thank you for your revision.
I look forward to further researches on the application of the mat in the future.
Reviewer 2 Report
Now, the revised version of the manuscript is acceptable for publication.
Reviewer 3 Report
Dear authors I found the revised paper suitable for publication.